# Synthesis, Characterization, Catalytic Activity, and DFT Calculations of Zn(II) Hydrazone Complexes

**DOI:** 10.3390/molecules25184043

**Published:** 2020-09-04

**Authors:** Temiloluwa T. Adejumo, Nikolaos V. Tzouras, Leandros P. Zorba, Dušanka Radanović, Andrej Pevec, Sonja Grubišić, Dragana Mitić, Katarina K. Anđelković, Georgios C. Vougioukalakis, Božidar Čobeljić, Iztok Turel

**Affiliations:** 1Faculty of Chemistry, University of Belgrade, Studentski trg 12–16, 11000 Belgrade, Serbia; adejumo.temiloluwa@gmail.com (T.T.A.); kka@chem.bg.ac.rs (K.K.A.); 2Laboratory of Organic Chemistry, Department of Chemistry, National and Kapodistrian University of Athens, Panepistimiopolis, 15771 Athens, Greece; nitzouras@gmail.com (N.V.T.); leozor888@gmail.com (L.P.Z.); 3Center for Chemistry, ICTM, University of Belgrade, Njegoševa 12, P.O. Box 815, 11001 Belgrade, Serbia; radanovic@chem.bg.ac.rs (D.R.); sonja.grubisic38@gmail.com (S.G.); 4Faculty of Chemistry and Chemical Technology, University of Ljubljana, Večna pot 113, 1000 Ljubljana, Slovenia; Andrej.Pevec@fkkt.uni-lj.si; 5Innovation Centre of Faculty of Chemistry, University of Belgrade, Studentski trg 12–16, 11000 Belgrade, Serbia; dmitic@chem.bg.ac.rs

**Keywords:** Girard’s T reagent, hydrazone ligand, Zn(II) complexes, XRD, ketone-amine-alkyne coupling reaction, catalysis, DFT calculation

## Abstract

Two new Zn(II) complexes with tridentate hydrazone-based ligands (condensation products of 2-acetylthiazole) were synthesized and characterized by infrared (IR) and nuclear magnetic resonance (NMR) spectroscopy and single crystal X-ray diffraction methods. The complexes **1**, **2** and recently synthesized [Zn**L^3^**(NCS)_2_] (**L^3^** = (*E*)-*N*,*N*,*N*-trimethyl-2-oxo-2-(2-(1-(pyridin-2-yl)ethylidene)hydrazinyl)ethan-1-aminium) complex **3** were tested as potential catalysts for the ketone-amine-alkyne (KA^2^) coupling reaction. The gas-phase geometry optimization of newly synthesized and characterized Zn(II) complexes has been computed at the density functional theory (DFT)/B3LYP/6–31G level of theory, while the highest occupied molecular orbital and lowest unoccupied molecular orbital (HOMO and LUMO) energies were calculated within the time-dependent density functional theory (TD-DFT) at B3LYP/6-31G and B3LYP/6-311G(d,p) levels of theory. From the energies of frontier molecular orbitals (HOMO–LUMO), the reactivity descriptors, such as chemical potential (*μ*), hardness (*η*), softness (*S*), electronegativity (*χ*) and electrophilicity index (*ω*) have been calculated. The energetic behavior of the investigated compounds (**1** and **2**) has been examined in gas phase and solvent media using the polarizable continuum model. For comparison reasons, the same calculations have been performed for recently synthesized [Zn**L^3^**(NCS)_2_] complex **3**. DFT results show that compound **1** has the smaller frontier orbital gap so, it is more polarizable and is associated with a higher chemical reactivity, low kinetic stability and is termed as soft molecule.

## 1. Introduction

Hydrazone ligands are one of the most important classes of flexible and versatile polydentate ligands which show very high efficiency in chelating various metal ions [1,2,3,4,5,6,7,8,9,10,11,12,13]. The coordination behavior of hydrazones is known to depend on the pH of the medium, the nature of the substituents and on the position of the hydrazone group relative to other moieties [2,3,4]. Moreover, deprotonation of the –NH group, which is readily achieved in the complexed ligand in particular, results in the formation of tautomeric anionic species (=N–N^−^–C=O or =N–N=C–O^−^), having different coordination properties. Our interest in metal complexes with hydrazone-based ligands is partly due to their potential applications as catalysts [5,6] and molecular magnets [7]. Hydrazone ligands, obtained from Girard’s T reagent (trimethylammoniumacetohydrazide chloride) and 2-acetylpyridine or 2-quinolinecarboxaldehyde (**HL^3^**Cl = (*E*)-*N*,*N*,*N*-trimethyl-2-oxo-2-(2-(1-(pyridin-2-yl)ethylidene)hydrazinyl)ethan-1-aminium chloride and **HL^4^**Cl = (*E*)-*N*,*N*,*N*-trimethyl-2-oxo-2-(2-(quinolin-2-ylmethylene)hydrazinyl)ethan-1-aminium chloride) are tridentate ligands which show the potential to form mononuclear and binuclear structures with metal ions [3,4,6,8,9,10]. In the mononuclear complexes (e.g., [Zn**L^3^**(NCS)_2_] (**3**) [8], [Cd**HL^3^**(NCS)_2_(SCN)] [8], [Zn**L^4^**(N_3_)_2_] [9], [Zn**L^4^**(NCO)_2_] [9], [Zn**L^4^**(N_3_)_1.65_Cl_0.35_] [10], [Cd**L^4^**(NCO)_1.64_Cl_0.36_] [10] and [Co**HL^4^**(N_3_)_3_] [4]) with one hydrazone ligand being coordinated in a tridentate fashion to the central metal ion, the vacant coordination sites are filled by monodentate ligands (pseudohalides/halides) and in some cases by counter anions of starting metal salts [11]. In the other type of mononuclear structures two hydrazone ligand molecules occupy the coordination sphere of the central metal ion forming an octahedral coordination geometry (e.g., [Co**L^3^**_2_][Co(NCS)_4_]BF_4_ [12] and Zn(II) complex with the 2-acetylthiazole (N4)-phenylthiosemicarbazone ligand (CCDC 1059843) [13]). In the binuclear [M_2_(**L^4^**)_2_(*μ*_1,1_-N_3_)_2_(N_3_)_2_]⋅H_2_O⋅CH_3_OH (M = Ni(II) and Co(II)) [3,4] and [Cu_2_(**L^3^**)_2_(*μ*_1,1_-N_3_)_2_](ClO_4_)_2_ [6] complexes, the hydrazone-based ligands show the same tridentate *N*,*N*,*O*-coordination mode forming the octahedral structure in the former two complexes cooperatively with two bridging and one terminal azide anion per metallic center and the square pyramidal in the latter with two bridging azides.

On the other hand, the propargylamines are a unique family of organic compounds, which has received ample attention by the wider scientific community [14,15]. The profound interest surrounding these compounds is partly due to the bioactive nature of certain members of their family [14,15,16,17]. Furthermore, propargylic amines are frequently encountered as intermediates in organic synthesis, providing facile access to a variety of structurally complex organic compounds [14,15]. Among these compounds, the subgroup of tetrasubstitutedpropargylamines is particularly interesting, as it comprises the least studied family of propargylamines. The most straightforward approach towards such molecules is the ketone-amine-alkyne (KA^2^) multicomponent coupling reaction, for which a significant number of catalytic systems has been reported during the past decade [18,19,20,21,22,23,24,25,26,27,28]. As part of the work of some authors focusing on sustainable organic transformations, multicomponent reactions and sustainable metal catalysis [19,20,21,22,23,24,25,26,27,28,29,30,31,32,33], the first zinc-based homogeneous catalytic system for the KA^2^ coupling was disclosed very recently [34]. Since the use of ligands in these catalytic systems is rare, we were interested in testing well-defined zinc complexes as potential catalysts for the reaction operating under air.

Herein, we report the synthesis and characterization of two new Zn(II) complexes [Zn**L^1^**(NCS)_2_]⋅2H_2_O (**1**) and [Zn(**L^2^**)_2_] (**2**)with the condensation products of 2-acetylthiazole and trimethylammoniumacetohydrazide chloride (Girard’s T reagent) and thiosemicarbazide, respectively. The structures of **1** and **2** were characterized by infrared (IR) and nuclear magnetic resonance (NMR) spectroscopy and single crystal X-ray diffraction methods. The catalytic activity of **1** and **2** and the previously characterized [Zn**L^3^**(NCS)_2_]⋅0.5MeOH [8] complex **3** has been evaluated in the ketone-amine-alkyne (KA^2^) coupling reaction. The experimental studies on the catalytic activities of **1**, **2** and recently synthesized [Zn**L^3^**(NCS)_2_]⋅0.5MeOH [8] (**3**) complex have been accompanied by density functional theory (DFT) calculations. We used B3LYP, a density functional theory-based approach, to describe the structures and molecular properties of the compounds of interest. The global reactivity descriptors, namely, chemical potential (*μ*), hardness (*η*), softness (*S*), electronegativity (*χ*), and electrophilicity index (*ω*) have been evaluated from highest occupied molecular orbital (HOMO) and lowest unoccupied molecular orbital (LUMO) energy values.

## 2. Results and Discussion

### 2.1. Synthesis

The ligand (**HL^1^**Cl), (*E*)-*N*,*N*,*N*-trimethyl-2-oxo-2-(2-(1-(thiazol-2-yl)ethylidene)hydrazinyl)ethan-1-aminium chloride*,* was obtained from the condensation reaction of 2-acetylthiazole and Girard’s T reagent. By the reaction of ligand **HL^1^**Cl with Zn(OAc)_2_∙4H_2_O and NH_4_SCN in excess in solvent mixture of water/methanol, mononuclear Zn(II) complex **1**, with the composition [Zn**L^1^**(NCS)_2_]⋅2H_2_O, was obtained (Scheme 1a). In complex **1**, Zn(II) is pentacoordinated with the thiazole nitrogen, the azomethine nitrogen, and the carbonyl oxygen atoms from the deprotonated hydrazone ligand and two thiocyanate ligands coordinated through the nitrogen atoms. Reaction of ligand **HL^2^** ((*E*)-2-(1-(thiazol-2-yl)ethylidene)hydrazine-1-carbothioamide) with Zn(BF_4_)_2_∙6H_2_O and NaN_3_ in a 1:1:2.3 molar ratios, in a mixture of acetonitrile/water, gives mononuclear Zn(II) complex **2**, with composition [Zn(**L^2^**)_2_] (Scheme 1b). Two deprotonated ligands **L^2^** are coordinated to Zn(II) ion through the thiazole nitrogen, the imine nitrogen, and the thiolate sulfur atoms, forming a distorted octahedral complex. Complex **3**, with composition [Zn**L^3^**(NCS)_2_]⋅0.5MeOH was previously reported [8] and has a similar structure to complex **1** (Scheme 1c).

### 2.2. Crystal Structures of [Zn**L^1^**(NCS)_2_]⋅2H_2_O (**1**) and [Zn(**L^2^**)_2_] (**2**) Complexes

The molecular structure of **1** is shown in Figure 1. Selected bond distances and angles are given in Table 1. The neutral complex [Zn**L^1^**(NCS)_2_] crystallizes as dihydrate in the triclinic crystal system with space group *P*−1. In **1**, Zn1 has fivefold coordination with tridentate ligand **L^1^** and two nitrogen atoms (N5, N6) from thiocyanate ligands. **L^1^** is coordinated to Zn1 in the zwitterionic form through NNO-set of donor atoms forming two fused five-membered chelate rings (Zn–N–C–C–N and Zn–N–N–C–O). The dihedral angle of nearly 4.0° between two five-membered chelate rings shows the non-coplanar nature of metal-ligand system. Generally, the distortion in the five coordinated systems is described by an index of trigonality *τ* = (*β* − *α*)/60, where *β* is the greatest basal angle and *α* is the second greatest angle [35]. The parameter *τ* is 0 for regular square based pyramidal forms and 1 for trigonal bipyramidal forms. The *τ* value of 0.36 calculated for **1**, indicates that the irregular coordination geometry about Zn1 is 36% trigonally distorted square-based pyramidal. The greatest basal angles O1−Zn1−N1 and N2−Zn1−N5 are 149.20(7)° and 127.42(10)°, respectively. The Zn1 is lifted out of the plane of the four in-plane ligand atoms (O1, N1, N2 and N5) by a distance *ρ* of 0.6091(3) Å. The complex **1** shows slightly greater degree of trigonal distortion from ideal square-based pyramidal configuration, in comparison with the other five-coordinate Zn(II) complexes with *N*-heteroaromaticmonohydrazones of Girard’s T reagent and pseudohalide or halide ligand (N_3_^−^, NCO^−^, NCS^−^ or Cl^−^) as monodentate for which the calculated *τ* values are in the range 0.31–0.34 (complexes **3**–**6**, Appendix A) [8,9,10]. The *τ* values for complexes **1**, **3**–**6** fit into the range of values obtained for related Zn(II)complexes [36,37,38,39,40] (complexes **7**–**15**, Appendix A). The Zn(II) ion in **1** is more strongly bound to the imine nitrogen atom of the ligand **L^1^** than to the 1,3-thiazole nitrogen, as indicated by the Zn1–N2, 2.058(2) Å and Zn1–N1, 2.212(2) Å bond lengths (Table 2). Similar to this, in analogous Zn(II) complexes with Girard’s T hydrazone-based ligands and N_3_^−^, NCO^−^, NCS^−^ or Cl^−^ as monodentate ligands [8,9,10] the Zn−N(imine), 2.049(3)–2.088(6) Å bond lengths are shorter than Zn–N(pyridine/quinoline), 2.206(6)/2.240(2)–2.344(2) Å bond lengths, indicating that Zn(II) ion is more tied to the imine nitrogen atom than to the pyridine or quinoline nitrogen atoms. In complexes **1**, **3**–**6**, the Zn−O(enolic) bond distances are in the range from 2.146(5) to 2.222(2) Å. The shortest Zn−O(enolic) bond length is found in the [Zn**L^3^**(NCS)_2_]⋅0.5MeOH [8] complex and the longest in [Zn**L^4^**(N_3_)_1.65_Cl_0.35_] [10] (**L^3^** = condensation product of 2-acetylpyridine and trimethylammoniumacetohydrazide chloride and **L^4^** = condensation product of. 2-quinolinecarboxaldehyde and trimethylammoniumacetohydrazide chloride). The thiocyanate ligands in **1** are coordinated to Zn(II) ion in bent mode, with Zn−N−C angles of 168.5(3) and 171.7(2)°. In the crystals of **1** the complex molecules [Zn**L^1^**(NCS)_2_] are assembled into supramolecular layers parallel with the (0 0 1) lattice plane by means of moderate O1W–H1W⋅⋅⋅O1 and weak C8–H8A⋅⋅⋅S2 and C10–H10B⋅⋅⋅O1W intermolecular hydrogen bonds [41] (Appendix A). In addition, the solvent water molecules O1W and O2W assists in joining the neighboring layers related by the center of symmetry by means of weak intermolecular hydrogen bonds C–H⋅⋅⋅OW (Appendix A). The shortest separations of 4.467(2) and 4.666(2) Å between the centers of gravity of the 1,3-thiazole rings are observed along the [−1 1 0] direction.

The molecular structure of **2** is shown in Figure 2. Selected bond distances and angles are given in Table 1. The neutral complex molecule [Zn(**L^2^**)_2_] crystallizes in the monoclinic crystal system with space group *P*2_1_/*c*. In complex **2**, two deprotonated ligand molecules **L^2^** coordinate the Zn(II) ion in a meridional fashion, forming a distorted octahedral complex by chelation through two NNS donor atom sets. Each ligand coordinates to metallic center through thiazole nitrogen, imine nitrogen and thiolate sulfur atoms. The tridentate coordination of each ligand implies the formation of two fused five-membered chelate rings Zn–N–C–C–N and Zn–N–N–C–S. The chelate rings (Zn1–N5–C9–C10–N6 and Zn1–N6–N7–C12–S4) are nearly coplanar, while the other pair (Zn1–N1–C3–C4–N2 and Zn1–N2–N3–C6–S2) deviates significantly from coplanarity, as indicated by the dihedral angles of 2.2° and 7.1°, respectively. In addition, the two chelation planes comprising the atoms N–N–S–Zn are practically perpendicular (dihedral angle = 89.7°). The octahedral complex molecule of **2** is comparable with the Zn(II) complex containing a similar ligand (2-acetylthiazole (N4)-phenylthiosemicarbazone) (CSD refcode KUMPEP) [13], although the latter is much more distorted due to the presence of the phenyl group at the terminal nitrogen atom of the thiosemicarbazone ligand, as evidenced by the smaller dihedral angle between chelation planes (N–N–S–Zn) compared to that observed in **2** (83.9° vs. 89.7°). One of the measures of the octahedral strain is average Δ*O_h_* value, defined as the mean deviation of 12 octahedral angles from ideal 90°. The complex **2** shows less octahedral strain in comparison to that observed in analogous Zn(II) complex with 2-acetylthiazole (N4)-phenylthiosemicarbazone. The calculated Δ*O_h_* values are 10° for the former and 12° for the latter complex. The mean Zn–**L** bond lengths (Zn–N_1,3-thiazole_ 2.2525 Å, Zn–S_thiolate_ 2.4313 Å and Zn–N_imine_ 2.148 Å) observed in complex **2** are similar to those found in its structural analogue (Zn–N_1,3-thiazole_ 2.2310 Å, Zn–S_thiolate_ 2.4331 and Zn–N_imine_ 2.1877Å).

In the crystals of complex **2,** molecules self-assemble within the layer parallel with the (1 0 0) lattice plain by means of intermolecular hydrogen bonds between terminal NH_2_ groups (N4 and N8) serving as hydrogen bond donors and thiolate sulfur atoms S4 at 1 − *x*, −1/2 + *y*, ½ − *z* and S2 at 1 − *x*, 2 − *y*, −*z* serving as acceptors (Appendix A). The complex molecules belonging to neighboring layers are linked through weak π⋅⋅⋅π interactions involving heteroaromatic 1,3-thiazole rings to form a 3D supramolecular structure (Appendix A). In addition, the molecules of **2** are linked along *a* crystallographic axis by weak C_aromatic_–H⋅⋅⋅N_hydrazone_ contacts.

### 2.3. Evaluation of the Zinc Complexes’ Catalytic Activity in the KA^2^ Coupling Reaction

We chose cyclohexanone, pyrrolidine and phenylacetylene as a model substrate triad. A promising result was obtained when complex **1** was used in 10 mol% loading in toluene, affording the product in 85% isolated yield after 16 h (Entry 1, Table 2). As expected, when ligand **HL^1^**Cl was used as a possible catalyst in a control experiment, the desired propargylamine was not formed. Complex **3** led to a 67% yield under the same conditions, while complex **2** also displayed moderate catalytic activity (Entries 3 and 5, Table 2), suggesting that the zinc center is no longer fully coordinated under the reaction conditions. Removing the solvent while reducing the temperature and catalyst loading also led to moderate yields in the cases of both **1** and **3** (Entries 6–8, Table 2), while using MgSO_4_ as a water-scavenging additive, in combination with an increase in temperature, led to the highest yield, when complex **1** was used in 5 mol% loading (Entry 9, Table 2). Under the same conditions, complex **3** led to moderate yield, while reducing the reaction time to 3 h led to incomplete conversion and low yield (Entries 10 and 11 respectively, Table 2), suggesting that the reaction conditions outlined in entry 9 of Table 2 were optimal. Of note, when taking into account the reactivity of simple zinc salts, complex **1** performs comparably well in this reaction. However, lower catalyst loading is required under the conditions described herein, while, in the case of zinc acetate, 10 mol% was essential in order to reach yields above 90%, in combination with dry/inert conditions.

Several substrate combinations were coupled under the aforementioned conditions, as shown in Scheme 2. Piperidine led to compound **4b** in high yield, as was the case in the parent, Zn-based, ligand-free system and the more recently reported Mn-based system [34,42]. Propargylamine **4c** was obtained in moderate yield, while using a linear ketone in combination with pyrrolidine afforded compound **4d** in 72% yield. Propargylamine **4e**, bearing an ester moiety that can be used for further functionalization, was synthesized in good yield, while the primary amine-derived compound **4f** was also successfully synthesized, albeit in moderate yield because of the stability of the intermediate imine. When the steric bulk of the linear ketone was increased, the yield dropped significantly, highlighting the crucial effect of steric hindrance in the outcome of this reaction (compound **4g**). When an aliphatic alkyne was used in combination with *N*-phenylpiperazine, propargylamine **4h** was obtained in 37% isolated yield. In order to assess the effect of a less functionalized aliphatic alkyne, 1-octyne was used and compound **4i** was isolated in 70% yield. Finally, cyclopentanone was chosen as a coupling partner and, as anticipated based on known reactivity trends, compound **4j** was obtained in moderate yield [34,42]. Overall, complex **1** allows for lower catalyst loading when compared to simple zinc salts and is more robust under harsh, ambient conditions [34]; however, the limitations of this coupling reaction and the generally observed trends regarding substrate scope persist in this case as well.

### 2.4. Density Functional Theory (DFT) Optimized Structures and Highest Occupied Molecular Orbital-Lowest Unoccupied Molecular Orbital (HOMO-LUMO) Analysis

In order to calculate the ground-state geometries of the complexes, DFT calculations of [Zn**L^1^**(NCS)_2_] (**1**) and [Zn(**L^2^**)_2_] (**2**), as well as [Zn**L^3^**(NCS)_2_] (**3**) complexes have been performed, as described below. DFT calculations predict five-fold coordination for both [Zn**L^1^**(NCS)_2_] and [Zn**L^3^**(NCS)_2_] complexes with tridentate ligands **HL^1^**Cl and **HL^3^**Cl and two nitrogen atoms from thiocyanate ligands (Figure 3), thereby supporting the experimental X-ray diffraction (XRD) results. In complex **2** DFT results show that two tridentate ligand molecules **L^2^** coordinate the Zn(II) ion through thiazole nitrogen, imine nitrogen and thiolate sulfur atoms, forming an octahedral complex with four fused five-membered chelate rings, in agreement with experimental data. Selected bond lengths and values of valence angles are summarized in Appendix A. The calculated geometric parameters of mixed ligand complexes are compared with the X-ray diffraction structures and show good agreement.

The HOMO-LUMO energies of the complexes provide information about energetic behavior and stability of the complexes. The energy gap between HOMO and LUMO, determines reactivity and kinetic stability of molecules [43,44,45]. The chemical hardness (*η*) is a good indicator of the chemical stability. The molecules having a large energy gap are known as hard and having a small energy gap are known as soft molecules. The soft molecules are more polarizable than the hard ones because they need little energy for excitation [46,47]. The chemical potential (*μ*), hardness value (*η*), softness (*S*), electronegativity (*χ*) and electrophilicity index (*ω*) of molecules are formulated by the equations [47]:*μ* = −(−E_HOMO_ − E_LUMO_)/2,(1)
*η* = (−E_HOMO_ + E_LUMO_)/2,(2)
*S* = 1/2*η*,(3)
*χ* = (−E_HOMO_ − E_LUMO_)/2,(4)
*ω* = *μ^2^*/*2η*,(5)
where E_HOMO_ and E_LUMO_ are the energies of the HOMO and LUMO orbitals. The negative chemical potential indicates complex to be stable in such a way that does not decompose spontaneously into its elements. Hardness measures the resistance to change in the electron distribution in a molecule.

The HOMO-LUMO energy calculations were performed within the time-dependent density functional theory (TD-DFT) approach at the B3LYP/6-31G level of theory in vacuum and toluene. This functional has been employed with a great success in reactivity studies, with a good compromise between accuracy and computational cost [48]. To examine the basis set dependence of the DFT HOMO and LUMO energies, we also performed TD-DFT calculations on the investigated systems using the B3LYP functional with a larger basis set such as 6-311G(d,p). Results are presented in the Appendix A. We obtained small differences between the HOMO and LUMO energies calculated at B3LYP level of theory by using the 6-31G and 6-311G(d,p) basis sets, ranging from 0.07 to 0.22eV. It has been already found that HOMO energies, negative values of LUMO energies and TD-DFT HOMO-LUMO gaps are generally less sensitive to the basis set [49].

The HOMO and LUMO and their energies were calculated to locate the high- and low-density regions in all complexes and are shown in Figure 4. The HOMOs of [Zn**L^1^**(NCS)_2_] (**1**) and [Zn**L^3^**(NCS)_2_] (**3**) complexes are delocalized mainly at the linear monodentate ligands NCS^–^ and metal centers, whereas the LUMOs are delocalized on the planar ring of Schiff base in equatorial plane (Figure 4). The HOMOs of the complex [Zn(**L^2^**)_2_] (**2**) are delocalized mainly at the five-membered heterocyclic rings with terminal NH_2_ groups. The LUMOs of the complex [Zn(**L^2^**)_2_] (**2**) are delocalized at the five-membered heterocyclic chelate rings of the two tridentate ligands. The calculated values of reactivity descriptors of complexes are given in Table 3. The negative values of chemical potential (−3.886, −3.597 and −3.670 eV) show their stability suggesting that these do not undergo decomposition into their components. As shown in Table 3, the compound that has the lowest energy gap in comparison to the two other complexes is the compound **1** (Δ*E*gap is 2.167 eV in vacuum and 2.977 eV in toluene). This lower energy gap allows it to be the softest molecule. The magnitude of chemical hardness, supported by the HOMO–LUMO energy gap, for complexes **1**, **2** and **3** have been found to be: 1.083, 1.456, and 1.232 eV, respectively (Table 3). Chemical hardness (softness) value of complex **1** is lower (greater) among all the investigated complexes, both in the gas phase and toluene. Hence, complex **1** is found to be more reactive than all the compounds which is in agreement with experimental catalytic data. The compound that has the lowest LUMO energy is the compound 1 (*E* = −2.803 eV) which signifies that it can be the best electron acceptor [50]. Besides, the electrophilicity index values *ω* given in Table 3 for complexes (6.971, 4.443 and 5.466 eV, respectively) related to chemical potential and hardness indicate that compound **1** is the strongest electrophile among all compounds. Compound **1** possesses a higher electronegativity value (*χ* = 3.886 eV) than all compounds, a characteristic that could explain its superior activity in catalysis, when compared to the other complexes evaluated herein [34]. Results were confirmed by using another DFT model denoted as BVP86/6-311G(d,p) with the lowest HOMO-LUMO energy gap for complex **1**. The differences between TD-DFT gaps calculated with selected different functionals are small. For instance, B3LYP and BVP86 predict relatively good HOMO and LUMO energies for investigated complexes with errors ranging from 0.56 to 0.73 eV.

## 3. Materials and Methods

2-Acetylpyridine (≥99%), 2-acetylthiazole (99%), thiosemicarbazide (99%) and Girard’s T reagent (99%) were obtained from Aldrich. IR spectra were recorded on a Nicolet 6700 Fourier transform infrared (FT-IR) spectrometer using the attenuated total reflectance (ATR) technique in the region 4000−400 cm^−1^ (vs-very strong, s-strong, m-medium, w-weak, bs-broad signal). ^1^H- and ^13^C-NMR spectra were recorded on a Bruker Avance Ultrashield 500 plus spectrometer (^1^H at 500 MHz; ^13^C at 125 MHz) at room temperature using TMS as internal standard in DMSO-*d*_6_, or on a Varian Mercury 200 MHz spectrometer (^1^H at 200 MHz; ^13^C at 50 MHz) or a Brucker Avance 400 MHz instrument (^1^H at 400 MHz; ^13^C at 101 MHz) using CDCl_3_ as the solvent. Chemical shifts are expressed in ppm (*δ*) values and coupling constants (*J*) in Hz. Elemental analyses (C, H, N and S) were performed by standard micro-methods using the ELEMENTARVario ELIII C.H.N.S.O analyzer. Molar conductivities were measured at room temperature (25 °C) on a digital conductivity-meter JENWAY-4009. GC/MS spectra were recorded with a Shimandzu R GCMS-QP2010 Plus Chromatograph Mass Spectrometer using a MEGAR (MEGA-5, F.T: 0.25 μm, I.D.: 0.25 mm, L: 30 m, Tmax: 350 °C, Column ID# 11475) column, using n-octane as the internal standard.

### 3.1. Synthesis of (E)-N,N,N-trimethyl-2-oxo-2-(2-(1-(thiazol-2-yl)ethylidene)hydrazinyl)ethan-1-aminium Chloride (**HL^1^**Cl)

The ligand **HL^1^**Cl was synthesized by the reaction of Girard’s reagent T (1.676 g, 10 mmol) and 2-acetylthiazole (1036 μL, 10 mmol) in water (20 mL). Reaction mixture was acidified with 3–4 drops of 2 M HCl and refluxed for 3 h. After cooling to the room temperature, white precipitate was filtered and washed with water. Yield: 2.54 g (92%). Anal. Calcd. (%) for C_10_H_17_N_4_OSCl: C, 43.40; H, 6.19; N, 20.24; S, 11.58. Found (%): C, 43.45; H, 6.21; N, 20.20; S, 11.52. IR (ATR, cm^−1^): 3387w, 3129w, 3092m, 3018m, 2955s, 1702vs, 1612w, 1550vs, 1487s, 1401m, 1300w, 1201s, 1135w, 976w, 945w, 914m, 787w, 748w, 684w, 585w, 552w. ^1^H-NMR (500 MHz, DMSO-*d*_6_), *δ* (ppm): 2.41, (s, 3H, C5-H); 2.53, (s, 3H, C5-H); 3.30, (s, 9H, C8-H); 3.34, (s, 9H, C8-H); 4.60, (s, 2H, C7-H); 4.82, (s, 2H, C7-H); 7.848, (d, 1H, *J*_C2-H/C3-H_ = 5 Hz, C2-H); 7.854, (d, 1H, *J*_C2-H/C3-H_ = 5 Hz, C2-H); 7.926, (d, 1H, *J*_C2-H/C3-H_ = 5 Hz, C3-H); 7.932, (d, 1H, *J*_C2-H/C3-H_ = 5 Hz, C3-H); 11.61, (s, 1H, N-H); 11.86, (s, 1H, N-H). ^13^C-NMR (125 MHz, DMSO-*d*_6_), *δ* (ppm): 13.9, (C5); 15.05, (C5); 53.6, (C8); 53.9, (C8); 63.0, (C7); 63.8, (C7); 123.3, (C2); 123.6, (C2); 143.9, (C3); 144.00, (C3); 147.00, (C4); 150.8, (C4); 161.2, (C1); 166. 8, (C1); 167.0, (C6); 167.3, (C6).

### 3.2. Synthesis of [Zn**L^1^**(NCS)_2_]⋅2H_2_O (**1**)

The Zn(II) complex **1** was synthesized by the reaction of Zn(OAc)_2_∙2H_2_O (75 mg, 0.30 mmol), ligand **HL^1^**Cl (83 mg, 0.30 mmol) and NH_4_SCN (60 mg, 0.78 mmol) in a solvent mixture of water/methanol (10/10 mL). The solution was refluxed for 4 h. After refrigeration of the reaction solution at −8 °C for two weeks pale yellow crystals suitable for X-ray analysis were formed. Yield: 0.11 g (83%). Anal. Calcd. (%) for C_12_H_20_N_6_O_3_S_3_Zn: C, 36.71; H, 5.13; N, 21.41; S, 24.51. Found (%): C, 36.55; H, 5.15; N, 21.27; S, 24.31. IR (ATR, cm^−1^): 3502s, 3383s, 3123m, 3055w, 2959w, 2088vs, 1610s, 1536s, 1475s, 1424s, 1402s, 1341m, 1290w, 1237w, 1151w, 1095w, 1064w, 1012w, 988w, 923w, 880w, 746w, 556w. ^1^H-NMR (500 MHz, DMSO-*d*_6_), δ (ppm): 2.54, (s, 3H, C5-H); 3.27, (s, 9H, C8-H); 4.20, (s, 2H, C7-H); 8.06, (d, 1H, *J*_C2-H/C3-H_ = 2.2 Hz, C2-H); 8.16, (d, 1H, *J*_C2-H/C3-H_ = 2.2 Hz, C3-H). ^13^C-NMR (125 MHz, DMSO-*d*_6_), *δ* (ppm): 15.4, (C5); 53.8, (C8); 66.7, (C7); 125.7, (C3); 134.3, (C9); 142.5, (C2); 147.3, (C4); 166.2, (C1); 171.7, (C6).

### 3.3. Synthesis of Ligand **HL^2^** (E)-2-(1-(thiazol-2-yl)ethylidene)hydrazine-1-carbothioamide

The ligand **HL^2^** was synthesized by the reaction of thiosemicarbazide and 2-acetylthiazole in water according to the previously described method [12]. Yield 1.82 g (91%). IR (ATR, cm^−1^): 3436s, 3248s, 3188s, 3099m, 3071m, 2983m, 2066w, 1648w, 1589s, 1510s, 1482s, 1452m, 1425s, 1365m, 1282m, 1166m, 1107m, 1069m, 1039m, 958w, 881w, 847w, 755w, 712w, 638w. Anal. Calcd. (%) for C_6_H_8_N_4_S_2_: C, 35.98; H, 4.03; N, 27.98; S, 32.02. Found (%): C, 35.74; H, 4.26; N, 27.88; S, 31.98. ^1^H-NMR (500 MHz, DMSO-*d*_6_), *δ* (ppm): 7.89, (C4-H, d *J^3^* = 5 Hz); 7.80, (C5-H, d, *J^3^* = 5 Hz); 2.43, (C7-H_3_, s); 10.67, (N-H, s); 8.53 and 7.69, (N-H_2_, bs). ^13^C-NMR (125 MHz, DMSO-*d*_6_), *δ* (ppm): 14.1, (C7); 144.7, (C2); 179.4, (C8); 167.5, (C6); 143.7, (C4); 123.1, (C5).

### 3.4. Synthesis of [Zn(**L^2^**)_2_] (**2**)

The Zn(II) complex **2** was synthesized by the reaction of Zn(BF_4_)_2_∙6H_2_O (72 mg, 0.20 mmol), ligand **HL^2^**Cl (40 mg, 0.20 mmol) and NaN_3_ (30 mg, 0.46mmol) in solvent mixture of acetonitrile/water (15/5 mL). The solution was refluxed for 4 h. After refrigeration of the reaction solution at –8 °C for one-week yellow crystals suitable for X-ray analysis were formed. Yield: 0.08 g (87%). Anal. Calcd. (%) for C_12_H_14_N_8_S_4_Zn: C, 36.16; H, 3.54; N, 28.12; S, 32.18. Found (%): C, 36.04; H, 3.56; N, 28.02; S, 32.07. IR (ATR, cm^−1^): 3571w, 3496w, 3436m, 3308vs, 3169s, 3109s, 2925w, 2255w, 2185w, 2078m, 1632m, 1589m, 1569m, 1496s, 1425vs, 1375vs, 1298s, 1198vs, 1162s, 1097s, 1029m, 882w, 783m, 726w, 677m, 644w, 596w, 547w, 480w. ^1^H-NMR (500 MHz, DMSO-*d*_6_), *δ* (ppm): 2.43, (s, 3H, C5-H); 7.45, (s, 2H, NH2); 7.88, (d, 1H, C2-H); 7.94, (d, 1H, C6-H). ^13^C-NMR (125 MHz, DMSO-*d*_6_), *δ* (ppm): 15.9, (C5); 122.5, (C3); 139.2, (C4); 142.4, (C2); 167.3, (C1); 181.7, (C6).

### 3.5. Synthesis of Ligand **HL^3^**Cl (E)-N,N,N-trimethyl-2-oxo-2-(2-(1-(pyridin-2-yl)ethylidene)hydrazinyl) ethan-1-aminium Chloride

The **HL^3^**Cl ligand was synthesized by the reaction of 2-acetylpyridine and Girard’s T reagent according to the previously described method [8]. Yield 2.36 g (87%). IR (ATR, cm^−1^): 3387w, 3127m, 3090m, 3049m, 3016m, 2950s, 1700vs, 1612w, 1549s, 1485m, 1400m, 1300w, 1253w, 1200s, 1153w, 1135m, 1095w, 1073m, 975w, 944w, 914m, 748w, 683w.

### 3.6. Synthesis of [Zn**L^3^**(NCS)_2_]⋅0.5CH_3_OH Complex (**3**)

The Zn(II) complex **3** was synthesized by the reaction of ligand **HL^3^**Cl, Zn(OAc)_2_·2H_2_O and NH_4_SCN according to the previously described method [8]. Yield 0.08 g (94%). IR (ATR, cm^−1^): 3030w, 2067vs, 1639w, 1620w, 1592w, 1566m, 1535s, 1464m, 1437m, 1397m, 1366m, 1339m, 1302m, 1200w, 1145w, 1074m, 1019m, 975w, 914w, 782w, 749 (w). *λ_M_* = 28 Ω^−1^cm^2^mol^−1^.

### 3.7. X-Ray Crystallography

The molecular structures of complexes **1** and **2** were determined by single-crystal X-ray diffraction. Crystallographic data and refinement details are given in Appendix A. The X-ray intensity data for **1** were collected at room temperature on a Nonius Kappa CCD diffractometer equipped with graphite-monochromator utilizing MoKα radiation (*λ* = 0.71073 Å). Data reduction and cell refinement was carried out using DENZO and SCALPACK [51]. Diffraction data for **2** were collected at room temperature with an Agilent SuperNova dual source diffractometer using an Atlas detector and equipped with mirror-monochromated MoKα radiation (*λ* = 0.71073 Å). The data were processed by using CrysAlis PRO [52]. All the structures were solved using SIR-92 [53] and refined against *F*^2^ on all data by full-matrix least-squares with SHELXL–2014 [54]. All non-hydrogen atoms were refined anisotropically. The water bonded hydrogen atoms in **1** was located in a difference map and refined with the distance restraints (DFIX) with O–H = 0.96 Å and with *U*_iso_(H) = 1.5*U*_eq_(O). All other hydrogen atoms were included in the model at geometrically calculated positions and refined using a riding model. Crystallographic data for the structures reported in this paper have been deposited with the CCDC 2021000 (for **1**) and 2021001 (for **2**). CCDC 2021000 and 2021001 contain the supplementary crystallographic data for this paper. These data can be obtained free of charge via http://www.ccdc.cam.ac.uk/conts/retrieving.html (or from the CCDC, 12 Union Road, Cambridge CB2 1EZ, UK; Fax: +44 1223 336033; E-mail: deposit@ccdc.cam.ac.uk).

### 3.8. Catalysis General Procedure

A Teflon sealed screw-cap pressure tube equipped with a stirring bar and a rubber septum or a screw-cap vial, was charged with x mol% of the catalyst and 0.5 eq. of the additive (MgSO_4_) unless otherwise noted. Under air, 0.5 mmol of the amine were added and the mixture was stirred until the solid was partially dissolved. 0.5 mmol of the alkyne were added and the mixture was stirred at room temperature. Finally, 0.5 mmol of the ketone were added and the reaction was allowed to stir in a preheated oil bath, for the appropriate time. After cooling to room temperature, ethyl acetate was added (2 × 5 mL) and the mixture was stirred for 5 min. The mixture was filtered through a short silica gel plug, in order to remove inorganic impurities, concentrated under vacuum and loaded atop a silica gel column. Gradient column chromatography with ethyl acetate/petroleum ether furnished the desired products. All products were characterized by ^1^H-NMR, and ^13^C{^1^H}-NMR which were all in agreement with the assigned structures and the data reported in the literature ([34,42] and references cited therein]).

#### Characterization Data for the Synthesized Propargylamines

*1-(1-(Phenylethynyl)cyclohexyl)pyrrolidine* (**4a**): Obtained as an orange/yellow oil in 91% yield (115 mg, 0.46 mmol). ^1^H-NMR (200 MHz, CDCl_3_) δ 7.44 (dd, *J* = 6.7, 3.1 Hz, 2H), 7.30 (m, 3H), 2.82 (t, *J* = 5.9 Hz, 4H), 2.13–1.94 (m, 2H), 1.91–1.41 (m, 12H, overlapping peaks). ^13^C{^1^H}-NMR (50 MHz, CDCl_3_) δ 131.7, 128.1, 127.6, 123.6, 90.3, 86.1, 59.3, 47.0, 37.8, 25.7, 23.5, 23.0.

*1-(1-(Phenylethynyl)cyclohexyl)piperidine* (**4b**): Obtained as a yellow oil in 87% yield (116 mg, 0.44 mmol). ^1^H-NMR (400 MHz, CDCl_3_) δ 7.47–7.36 (m, 2H), 7.28–7.23 (m, 3H), 2.70 (m, 4H), 2.11–2.05 (m, 2H), 1.83–1.34 (m, 14H, overlapping peaks). ^13^C{^1^H}-NMR (101 MHz, CDCl_3_) δ 131.9, 128.4, 127.8, 124.0, 90.6, 86.3, 59.7, 47.3, 35.8, 26.6, 25.8, 24.9, 23.2.

*1-(1-(p-Tolylethynyl)cyclohexyl)piperidine* (**4c**): Obtained as a yellow oil in 64% yield (91 mg, 0.32 mmol). ^1^H-NMR (400 MHz, CDCl_3_) δ 7.33 (d, *J* = 7.9 Hz, 2H), 7.10 (d, *J* = 7.9 Hz, 2H), 2.73 (s, 4H), 2.34 (s, 3H), 2.12–2.09 (m, 2H), 1.78–1.42 (m, 14H, overlapping peaks). ^13^C{^1^H}-NMR (101 MHz, CDCl_3_) δ 137.9, 131.7, 129.1, 120.6, 90.4, 86.7, 59.9, 47.3, 35.7, 26.4, 25.8, 24.7, 23.3, 21.5.

1*-(3-Methyl-1-phenylpent-1-yn-3-yl)pyrrolidine* (**4d**): Obtained as an orange/brown oil in 72% yield (82 mg, 0.36 mmol). ^1^H-NMR (400 MHz, CDCl_3_) δ 7.45–7.38 (m, 2H), 7.32–7.26 (m, 3H), 2.80 (t, *J* = 5.8 Hz, 4H), 1.84–1.79 (m, 4H), 1.74–1.65 (m, 2H), 1.42 (s, 3H), 1.05 (t, *J* = 7.5 Hz, 3H). ^13^C{^1^H}-NMR (101 MHz, CDCl_3_) δ 131.7, 128.2, 127.7, 123.5, 90.9, 84.7, 58.6, 47.8, 33.9, 25.1, 23.7, 8.9.

*1-(1-(Phenylethynyl)cyclohexyl)piperidine-4-carboxylate* (**4e**): Obtained as a yellow oil in 67% yield (114 mg, 0.34 mmol). ^1^H-NMR (200 MHz, CDCl_3_) δ 7.42 (dd, *J* = 6.7, 3.0 Hz, 2H), 7.35–7.23 (m, 3H), 4.13 (q, *J* = 7.1 Hz, 2H), 3.15 (d, *J* = 11.6 Hz, 2H), 2.42–2.18 (t, *J* = 11.0 Hz, 3H), 2.13–1.43 (m, 14H), 1.24 (t, *J* = 7.1 Hz, 3H). ^13^C{^1^H}-NMR (50 MHz, CDCl_3_) δ 175.4, 131.7, 128.3, 127.8, 123.6, 90.4, 86.2, 60.3, 59.0, 45.8, 41.6, 35.9, 28.9, 25.8, 22.9, 14.3.

*N-Octyl-1-(phenylethynyl)cyclohexanamine* (**4f**): Obtained as a yellow oil in 53% yield (83 mg, 0.27 mmol). ^1^H-NMR (200 MHz, CDCl_3_) δ 7.42 (dd, *J* = 6.7, 3.1 Hz, 2H), 7.33–7.23 (m, 3H), 2.79 (t, *J* = 7.1 Hz, 2H), 1.94 (d, *J* = 11.6 Hz, 2H), 1.74–1.07 (m, 20H), 0.88 (m, 3H). ^13^C{^1^H}-NMR (50 MHz, CDCl_3_) δ 131.6, 128.1, 127.7, 123.6, 93.3, 84.6, 55.2, 43.2, 38.1, 31.8, 30.5, 29.5, 29.3, 27.5, 25.9, 23.1, 22.7, 14.1.

*1-(3-Methyl-1-phenylhex-1-yn-3-yl)pyrrolidine* (**4g**): Prepared according to the general procedure and obtained as an orange/brown oil in 31% yield (37 mg, 0.16 mmol). ^1^H-NMR (200 MHz, CDCl_3_) δ 7.45–7.36 (m, 2H), 7.28–7.25 (m, 3H), 2.79 (t, *J* = 5.4 Hz, 4H), 1.85–1.74 (m, 4H), 1.73–1.47 (m, 4H), 1.43 (s, 3H), 0.95 (t, *J* = 7.1 Hz, 3H). ^13^C{^1^H}-NMR (50 MHz, CDCl_3_) δ 132.0, 128.4, 127.9, 123.8, 91.5, 84.7, 58.3, 48.0, 44.0, 26.1, 23.9, 18.0, 14.8.

*2-Methyl-4-(1-(4-phenylpiperazin-1-yl)cyclohexyl)but-3-yn-2-ol* (**4h**): Obtained as an orange solid in 37% yield (60 mg, 0.19 mmol). ^1^H-NMR (400 MHz, CDCl_3_) δ 7.26 (m, 2H), 6.93 (d, *J* = 8.6 Hz, 2H), 6.89–6.80 (m, 1H), 3.31–3.15 (m, 4H), 2.88–2.73 (m, 4H), 2.25 (br. s, 1H), 2.04–1.38 (m, 16H). ^13^C{^1^H}-NMR (50 MHz, CDCl_3_) δ 151.3, 129.2, 119.7, 116.0, 91.6, 81.9, 65.4, 58.2, 49.6, 46.1, 35.7, 32.1, 25.7, 22.9.

*1-(1-(Oct-1-yn-1-yl)cyclohexyl)piperidine* (**4i**): Prepared according to the general procedure and obtained as a yellow oil in 70% yield (96 mg, 0.35 mmol). ^1^H-NMR (400 MHz, CDCl_3_) δ 2.70–2.50 (m, 4H), 2.22 (t, *J* = 6.7 Hz, 2H), 1.93 (d, *J* = 11.2 Hz, 2H), 1.78–1.70–1.24 (m, 22H), 0.95–0.79 (m, 3H). ^13^C-NMR (101 MHz, CDCl_3_) δ 85.8, 80.7, 58.9, 47.0, 36.0, 31.4, 29.3, 29.3, 25.9, 24.9, 23.2, 22.7, 18.7, 14.1.

*1-(1-(Phenylethynyl)cyclopentyl)piperidine* (**4j**): Prepared according to the general procedure and obtained as a yellow oil in 67% yield (339 mg, 1.34 mmol). ^1^H-NMR (400 MHz, CDCl_3_) δ 7.41 (dd, *J* = 6.5, 3.2 Hz, 2H), 7.31–7.25 (m, 3H), 2.75–2.60 (m, 4H), 2.15–2.10 (m, 2H), 1.87–1.44 (m, 12H, overlapping peaks). ^13^C{^1^H}-NMR (101 MHz, CDCl_3_) δ 131.8, 128.3, 127.7, 123.9, 91.6, 85.4, 67.7, 50.4, 39.9, 26.3, 24.5, 23.5.

### 3.9. DFT Calculations

Density functional theory was used to determine the optimized geometries of complexes [Zn**L^1^**(NCS)_2_] (**1**) and [Zn(**L^2^**)_2_] (**2**). For comparison reasons the same calculations have been performed for [Zn**L^3^**(NCS)_2_] (**3**) complex [8]. All the DFT calculations were performed in gas phase with the Gaussian 09 [55] program at B3LYP/6–31G level [56,57,58,59] of theory. In order to evaluate the energetic behavior of complexes, the HOMO-LUMO energy calculations were performed within the TD-DFT approach using B3LYP/6-31G, B3LYP/6-311G (d,p) and BVP86/6-311G(d,p) methods in vacuum. In addition, HOMO-LUMO energy investigations were also conducted in toluene by using TD-DFT/PCM (polarizable continuum model) [60] calculations at B3LYP/6-31G level of theory. Calculations were carried on the PARADOX supercomputing facility [61].

## 4. Conclusions

In the reactions of Zn(II) with (*E*)-*N*,*N*,*N*-trimethyl-2-oxo-2-(2-(1-(thiazol-2-yl)ethylidene)hydrazinyl)ethan-1-aminium chloride (**H****L^1^**Cl) and (*E*)-2-(1-(thiazol-2-yl)ethylidene)hydrazine-1-carbothioamide (**H****L^2^**) in the presence of NH_4_SCN and NaN_3_, respectively, two different structural types of Zn(II) complexes were obtained and characterized by spectroscopic methods (IR and NMR) and single crystal X-ray diffraction methods. In **1** the Zn(II) ion is pentacoordinated with tridentate ligand **L^1^** and two thiocyanate anions forming a distorted square pyramidal structure. In **2** the Zn(II) ion is hexacoordinated with two **L^2^** molecules forming a distorted octahedral structure.

The DFT calculations of newly synthesized **1** and **2** complexes, as well as [Zn**L^3^**(NCS)_2_] (**3**) complex have been carried out for their structural determination, HOMO, LUMO study and to calculate reactivity descriptors. The lower kinetic stability and higher reactivity of complex **1** compared to the other two complexes have been found from the lower HOMO–LUMO energy gap value, in agreement with experimental data. The electrophilicity index value *ω* (6.971 eV in gas phase and 5.908 eV in toluene) indicates that compound **1** is the strongest electrophile than all investigated compounds. In addition, compound **1** possesses higher electronegativity value (*χ* = 3.886 eV) than all compounds. Therefore, it is the best electron acceptor and that feature can plausibly explain its better performance as a Lewis acid catalyst in the ketone-amine-alkyne coupling.

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
