# Peer review of "Synthesis, Characterization, Catalytic Activity, and DFT Calculations of Zn(II) Hydrazone Complexes"

_molecules, 2020, doi:10.3390/molecules25184043_

Round 1
Reviewer 1 Report
The manuscript presence a rather big volume of date spanning from complex syntheses, across their potential application, to theoretical calculation. Unfortunately, some of these topics have been touched only superficially.
The part regarding synthesis of zinc complexes is described in detail including characterization of the prepared compounds.
The second part, application of the prepared complexes in organic synthesis, is unfortunately cover inadequately. The scope of the reaction is demonstrated on 7 substrates only. This is definitely not sufficient to justify potential application of the complexes as future catalysts for this type of the reaction. Firstly, other cyclic ketones with various ring sizes should be tested. Secondly, a broader selection of various terminal alkynes should undergo screening as well. Thirdly, the same comment applies for amines.
The third part is devoted DFT calculation. This part is generally OK, but it should be discussed prior to the organic synthesis section.
In addition, talking about reactivity of individual complexes, any conclusions should be based on data obtained by the method of initial rates. It would be also helpful to know kinetic profiles of the reactions catalyzed by various complexes. These data would shed light on reactivity and robustness of the catalysts as well.
The authors should address the above raiosed issues in the revised version.
Reviewer 2 Report
This is a nice paper that presents a detailed study on the structures and catalytic properties of new zinc hydrazone complexes. The synthesis, spectroscopic and computational characterization were well conducted with care; the X-ray structures were also nicely depicted. I suggest minor revisions to be made before it could be accepted by this journal.
In the catalysis section, it would be better to conduct a control experiment as an example using ligand-free zinc salt and list the results in Table 2. In addition, to report the isolated yields of product 4b-h, the authors should provide their NMR spectroscopic data and scanned copy of their spectra in the SI, even though they are known compounds. The references for comparison of spectroscopic data should be cited as well.
Additional minor revision: in the experimental section for the synthesis of new compounds, gram yields should be given along with % yields.
Reviewer 3 Report
The authors describe the synthesis, characterization, DFT studies and their employment in a KA2 coupling reaction. The studies are in general very well conducted, and the described catalysts show high potential for the mentioned reaction. Overall, the method and accompanying studies are of general interest and deserve publication in Molecules. However, some minor issues have to be corrected and one major issue regarding the DFT calculations is of special concern. -) In page 3, “In complex 1, Zn(II) is pentacoordinated with the pyridine nitrogen”. Complex 1 does not contain such pyridine moiety -) In scheme 1, the numbers of the ligands, L1, L2, L3 should be added below the structures, for clarity -) In Table 2, during the optimization studies, the reaction in toluene seems to work better than the neat conditions, if I understand properly (entry 7 vs entry 1). At the same time, the presence of MgSO4 improves the results (entry 9 vs entry 1). Thus, have the authors tried the combination of MgSO4 in toluene? -) What is the yield of the background reaction, without ligands, just with ZnCl2 or ZnOAc2? Do they work? Reference 33 should be probably more conveniently commented. -) How can be explained the difference in yield and reactivity between 4b and 4c, being both substrates so similar? -) How can complex 2 be formed, containing 2 units of ligand L2, if a single equivalent is used? Is the conversion low? -)) The major concern refers to the use of a short basis set for the DFT studies on HOMO and LUMO energies. 6-31G in combination with B3LYP is probably not enough to obtain accurate results. At least 6-311G** or similar should be necessary. Otherwise, justify the use of such a short system. According to Musgrave (J. Phys. Chem. A 2007, 111, 1554-1561), Seferos (dx.doi.org/10.1021/ma4005023 in Macromolecules 2013, 46, 3879−3886) and others, especially LUMO energies are not well predicted by most DFT methods. So, care has to be taken and a comparison between at least two different methods has to be conducted.Author Response
Please see the attachment.

Round 2
Reviewer 1 Report
The authors have provided a detailed reply to the raised issues and comments.
a) It should be appreciated that they added two more examples into the scheme 2.
b) Nonetheless, it is necessary to play devil's advocate to some of the authors' comments.
A comment to the reply to the comment 1.
A comparison of the obtained data in the manuscript and those in ref. 33 provides evidence that the complex 1 (5 mol% load) is obviously catalytically more efficient then uisng simple Zn salts (20 mol% load). On the other hand, it is generally accepted that if one wants to demonstrate an improved or more efficient catalytic system (or a reaction), a number of examples demonstrating its superiority should (must) be higher then in original reports. Otherwise, he could be a general chemical community conviced about its superiority?
A comment to to the reply to the comment 3.
A comparison of catalytic efficiency of two catalysts (or any processes) should (must) not be based on a simple comparison of two values (yields).
Firstly, these values (yields) were obtained after isolation from the reaction mixture, hence they do not represent conversion(s). (Some amount of the product is always lost during isolation and this amount is not known.)
Secondly, it is not clear whether the course of the reaction is finished after 16 h or the catalytic cycle stoped earlier. Hnece, in this respect even a simple time profile of the course of the reaction (conversion/yeild vs. reaction time) would provide useful information on catalytic efficacy of the used complexes. However, it is missing.
Taking into the account reasons stated in the reply regarding execution of additional experiments as well as the fact that other two reviewers did not raise similar inquiries, no further action is required. Although it would be highly recommended to proceed with them in order to increase the future impact of the reported data.